# Rational Decision-Making Agent with Internalized Utility Judgment

## Abstract

Large language models (LLMs) have demonstrated remarkable advancements and have attracted significant efforts to develop LLMs into agents capable of executing intricate multi-step decision-making tasks beyond traditional NLP applications. Existing approaches to LLM-based decision-making predominantly build upon the external performance measure to guide the decision-making process. However, reliance on the external performance measure as prior is problematic in real-world scenarios, where such prior may be unavailable, flawed, or even erroneous. For genuine autonomous decision making for LLM-based agents, it is imperative to develop rationality from their posterior experiences to judge decisions independently. Central to the development of rationality is the construction of an internalized utility judgment, capable of assigning numerical utilities to each decision. In this work, we propose RADAGENT (Rational Decision-Making Agent), which fosters the development of its rationality through an iterative framework involving *Experience Exploration* and *Utility Learning*. Within this framework, Elo-based Utility Construction is devised to assign Elo scores to individual decision steps to judge their utilities via pairwise comparisons. Consequently, these Elo scores guide the decision-making process to derive optimal outcomes. Experimental results on the Game of 24, WebShop, and ToolBench dataset demonstrate RADAGENT's superiority over baselines, achieving over 10% improvement in Pass Rate on diverse tasks. It offers higher-quality solutions and reduces costs (ChatGPT API calls), highlighting its effectiveness and efficiency.

## 1 Introduction

Agent (Searle, 1969; Wooldridge & Jennings, 1995; Maes, 1994; Hendler, 1999), as the long-standing pursuit of artificial intelligence (AI), is expected to possess the ability to plan, make decisions, and take actions to accomplish complex tasks autonomously. As large language models (LLMs) have undergone rapid development, showcasing remarkable capabilities (OpenAI, 2022; 2023), many efforts have been devoted to developing LLM-based agent (Richards, 2023; Nakajima, 2023; age, 2023) to accomplish intricate multi-step decision-making tasks (Yao et al., 2022b; Hao et al., 2023a; Yao et al., 2023; Qin et al., 2023c) beyond traditional natural language processing (NLP) applications. Even with these strides, existing LLM-based agents require external performance measures to guide their decision-making process (Yao et al., 2023; Hao et al., 2023a; Besta et al., 2023; Sel et al., 2023). For instance, in Game of 24, which uses four numbers and basic arithmetic operations to obtain 24, a prompt (Yao et al., 2023) is heuristically designed to assess the possibility of each decision to reach 24 and then choose correct decisions accordingly. However, this manual-designed prompt may not provide accurate possibility, causing unreliable decision-making guidance. The reliance on the external performance measure as *prior* restricts the adaptability in real-world scenarios as such prior may be unavailable, flawed, or even erroneous.

When individuals make decisions, they not only rely on external measures but also draw upon their practical experience as *posterior* to form a sense of individual rationality. This rationality can be conceptualized as an internal ability to judge utility, which encompasses two fundamental properties (Kahneman & Tversky, 2000; Arrow, 1959; Plott, 1973): (1) *Completeness*: When presented with any two choices $A$ and $B$, individuals are capable of establishing a strict preference for one over the other ($A \geq B$ or $B \geq A$). (2) *Transitivity*: If an individual prefers $A$ to $B$ ($A \geq B$) and also prefers $B$ to $C$ ($B \geq C$), then it logically follows that the individual must prefer choice $A$ to choice

$C$ ($A \geq B \geq C$). Drawing on these two fundamental properties, individuals can assess the utilities of a given set of choices and select the one with the highest utility to attain the optimal outcome.

To this end, we propose RADAGENT (Rational Decision-Making Agent) which internalizes the utility judgment ability to achieve rationality for the agent. In RADAGENT, the internalized utility judgment is constructed based on an iterative framework: (1) **Experience Exploration**: Due to the complexity of real-world tasks, the solution space may be infinite, and it is challenging to find the optimal solution efficiently. The agent should explore potential decisions to find better solutions as many as possible for the utility learning. (2) **Utility Learning**: Given a series of solutions, the agent should make comparisons between them to learn their utilities. To learn a quantitative utility, we design Elo-based Utility Construction which assigns each decision with an Elo score to represent its utility as the quantitative judgment through a series of pairwise comparisons between any two solutions. After multiple comparisons, the Elo score converges to an accurate value representing its actual utility in achieving the task. Using the learned utilities as guidance, the exploration process focuses on discovering decisions with higher utilities. Consequently, the exploration of these enhanced decisions aids in further refining the utilities associated with each decision. Through the iterative utility judgment construction, RADAGENT can assess the numerical utility of explored decisions and then can judge the highest utility to derive the best solution with the best outcome.

To validate the effectiveness of our proposed RADAGENT approach, we implement it based on ChatGPT (OpenAI, 2022) and conduct extensive experiments on Game of 24 (Yao et al., 2023), WebShop (Yao et al., 2022a), and ToolBench dataset (Qin et al., 2023c), which contains intricate multi-step decision tasks involving diverse scenarios. Experimental results unequivocally demonstrate the superiority of our approach against several baselines by achieving over 10% improvements in Pass Rate to accomplish complex tasks. Moreover, extensive analyses show that our approach not only delivers superior solutions with higher quality but also achieves greater efficiency by reducing the number of ChatGPT API calls.

Our contributions are threefold:

- We propose RADAGENT, a rational decision-making agent that can construct its internal rationality to accomplish diverse real-world tasks, not relying on external performance measure.
- We devise Elo-based Utility Construction which can internalize the utility judgment for the agent by learning Elo scores for each decision, selecting the optimal solution with the highest utilities.
- Extensive experiments on the Game of 24, WebShop, and ToolBench dataset demonstrate the effectiveness and efficiency of our proposed method against representative methods, marking a significant step toward unleashing the autonomous decision-making capability of LLMs.

## 2 PRELIMINARY

**Elo Rating System** The Elo rating system (Elo, 1967), commonly used in competitive contexts offers a numerical estimation of the skill levels of players. It represents the skill levels of players by Elo scores and assesses the Elo scores through a series of one-to-one competitions. It assumes that each player's performance follows a Gaussian distribution ($x \sim \mathcal{N}(\mu, \sigma)$) and each comparison of two players is actually comparing between two samples from their Gaussian distributions. Through multiple comparisons, we can approximate their true skill levels by estimating their Elo scores.

Given two players $x$ and $y$, their Elo scores are denoted as $v_x$ and $v_y$, respectively. The expected superiority of $x$ against $y$ is calculated as:

$$E_{x>y} = \frac{1}{1 + e^{-\frac{v_x - v_y}{r}}} \quad (1)$$

where $r$ is the Elo coefficient.

Next, we run a competition between them to find the actual winner. We denote the competition result as $R_{x>y}$:

$$R_{x>y} = \begin{cases} 1, \text{if } x \text{ win}, \\ 0, \text{if } y \text{ win}, \\ 0.5, \text{otherwise} \end{cases} \quad (2)$$

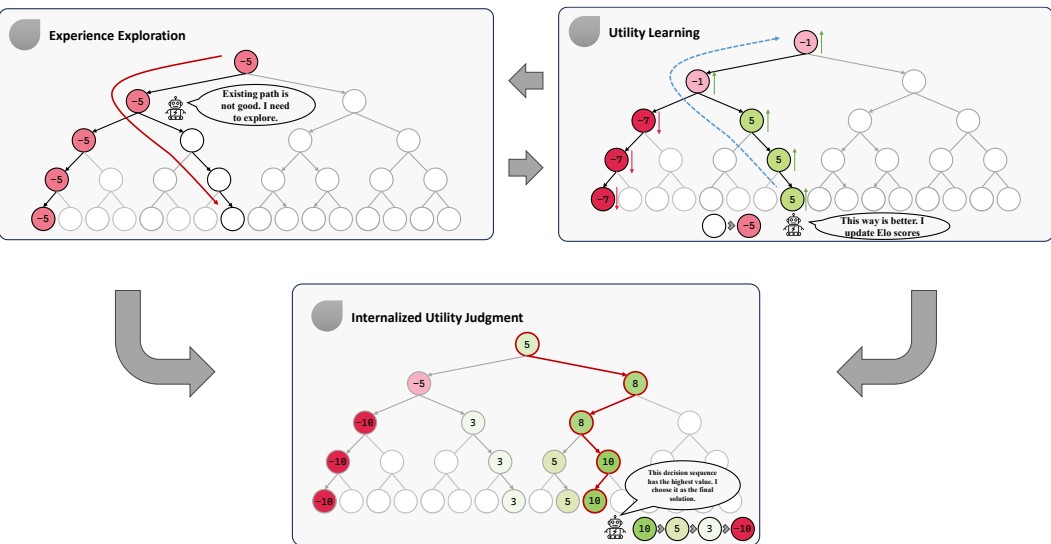

Figure 1: Illustration of the iterative Experience Exploration and Utility Learning phase to derive the final optimal solution.

We then update their Elo score accordingly:

$$v_x = v_x + K * (R_{x>y} - E_{x>y})$$
$$v_y = v_y + K * (R_{y>x} - E_{y>x})$$

(3)

where $K > 0$ is a hyper-parameter to control the update step.

**Markov Decision Process**  The decision-making process within agents is usually formulated as a Markov decision process (MDP). Given a human instruction $Q$, agents are tasked with generating a decision sequence $t = \{s_0, a_1, s_1, \cdots, s_N\}$ to accomplish $Q$. Here, $\{s_i\}_{i=0}^N$ represents the decision states, $s_0$ is the initial state, $s_N$ is the final state which means that agents accomplish the instruction, and $\{a_i\}_{i=1}^T$ denotes the actions taken by agents during the decision-making process. At each step in the MDP framework, agents decide to take action $a_i \sim P(a_i|s_i)$ based on the current state and subsequently arrive at the next state $s_{i+1} \sim P(s_{i+1}|a_i, s_i)$. Thus, we denote a decision step as $d_{i+1} = (s_i, a_i, s_{i+1})$. To make sequential decisions toward accomplishing $Q$ autonomously, agents need to identify the utility $v_{i+1} = V(d_{i+1})$ of each decision step $d_{i+1}$ and select those decision steps with a higher value that holds the promise of yielding the most promising outcomes, ultimately leading to the derivation of the final decision sequence that fulfills the requirements of $Q$.

## 3  METHODOLOGY

Our RADAGENT aims to find the decision sequence with the highest utility to accomplish complex instructions autonomously. It contains two principal phases to internalize the utility judgment:

- *Experience Exploration*: The agent takes actions sequentially to form a decision sequence toward a feasible solution.
- *Utility Learning*: The agent makes judgments among decision sequences to assess the utility (i.e., Elo scores) of existing decision steps.

These two phases work in an iterative fashion, reinforcing each other's outcomes (as shown in Figure 1). In the experience exploration phase, the agent explores more potential decision sequences to find better solutions, which can encourage agents to learn the actual and accurate utility of each decision step. In the utility learning phase, the agent re-calculates the Elo score of each decision step with the newly explored decision sequence to learn the utility of each decision, and the learned utilities serve as a dynamic guide, steering subsequent experience exploration toward more promising

and superior solutions. By iteratively cycling through these intertwined phases, the agent progressively evolves toward an optimal decision sequence with the highest utility to address instructions.

## 3.1 EXPERIENCE EXPLORATION

In RADAGENT, each experience exploration benefits from the previous exploration history based on Elo-based Utility Construction (§ 3.2). When exploring a new decision sequence, agents will select a decision step with a higher Elo score to explore further. Specifically, in RADAGENT, each decision step is assigned an Elo score explicitly. A decision step with higher Elo scores means that it is more likely to accomplish the instruction and thus Elo scores are used to guide the decision exploration process. Given an intermediate decision step $d$, its subsequent decision steps are denoted as $\{d_1, d_2, \cdots, d_n\}$. Given their learned Elo scores $\{v_i\}_{i=1}^n$, the probability of choosing to explore can be modified as follows:

$$P(d_i) = \frac{\exp(\frac{v_i}{\tau})}{\sum_j \exp(\frac{v_j}{\tau})}, \ d_i \in \{d_1, d_2, \cdots, d_n\} \tag{4}$$

where $\tau$ refers to the temperature. Note that only exploring the known decisions may cause the local optimal solution. Therefore, we define a rejection decision step $\hat{d}$ with an initial Elo score $\hat{v}$ to represent that *"The agent decides to explore a new decision"*. We add this rejection decision step into the subsequent decision steps as $\{d_1, d_2, \cdots, d_n, \hat{d}\}$ when selecting:

$$P(d_i) = \frac{\exp(\frac{v_i}{\tau})}{\sum_j \exp(\frac{v_j}{\tau})}, \ d_i \in \{d_1, d_2, \cdots, d_n, \hat{d}\} \tag{5}$$

The complete experience exploration process begins from the initial state $s_0$ and chooses the subsequent decision steps iteratively based on Equation 5 in a top-down manner. When it chooses the rejection decision step $\hat{d}$, the agent will generate a new decision sequence starting from the intermediate step $d$. In the iterative experience exploration process, those potential decision steps will be explored thoroughly, until finding the optimal solution.

## 3.2 UTILITY LEARNING

As external performance measures may be unavailable, flawed, or even erroneous, the agent should resort to their internalized utility judgment ability to solve diverse tasks. To this end, we design an Elo-based Utility Construction, equipping the agent with the Elo rating system to provide a numerical utility to each decision step to guide the decision-making process.

The utility learning process (i.e., Elo score estimation process) is conducted in a bottom-up manner. It first adjusts the Elo scores of the final decision steps of each decision sequence via pairwise comparison and then updates the Elo scores of the intermediate decision steps gradually. Once a new decision sequence is generated in the experience exploration phase, the agent will self-judge the Elo scores of existing decision steps via pairwise comparison. Given the newly generated decision sequence $t_n$, we first assign all decision steps of $t_n$ with an initial Elo score. Then, we randomly select a decision sequence $t_i$ from existing decision sequences $\mathbb{T} = \{t_1, t_2, \cdots, t_{n-1}\}$ and use agents to compare $t_n$ with $t_i$ to judge which one has the superior performance. Since the LLM-based comparison is sensitive to the candidate order (Qin et al., 2023d; Chiang & Lee, 2023; Wang et al., 2023), we conduct comparisons twice with different orders.

$$R_{t_n > t_i} = \begin{cases} 1, \text{if } t_n \text{ win twice,} \\ 0, \text{if } t_i \text{ win twice,} \\ 0.5, \text{otherwise} \end{cases} \tag{6}$$

Getting the comparison result, we update the Elo scores of the final decision steps of $t_n$ and $t_i$ based on Equation 3. Next, we calculate the Elo scores of intermediate decision steps based on their subsequent decision steps. Specifically, given an intermediate decision step $d_i$, we calculate its Elo scores as follows:

$$v_i = \sum_{d_j \in \text{Child}(d_i)} (\alpha_j * v_j), \tag{7}$$

where $\text{Child}(d_i)$ refers to the set of the subsequent decision steps of $d_i$, $\alpha_j = \frac{\exp(v_j/\tau)}{\sum_k \exp(v_k/\tau)}$ is the normalized weight and $\tau$ is from Equation 5. By repeating the comparison via randomly sampling decision sequences, the Elo score of each decision step will converge to its expected value.

When guiding the experience exploration process, the Elo score of a decision step with a few number of Elo updates may not represent its real value accurately. Such a decision step cannot be fully trusted for exhaustive exploration. Hence, we adjust the temperature $\tau$ in Equation 5 based on the number of the Elo update. Let $M_d$ be the number of the Elo update of the decision step $d$. The temperature of $d$ is annealed as follows:

$$\tau_d = \tau_0 * \frac{1}{1 + \sqrt{\ln(M_d + 1)}} \tag{8}$$

where $\tau_0$ is the default temperature. With the growth of the number of Elo updates, the approximated Elo score converges to its real value. At this time, we tend to explore the most possible decision.

### 3.3 RATIONALITY CONSTRUCTION

After engaging in extensive experience exploration and utility learning, the agent internalizes the utility judgment to construct rationality. Rationality for an individual is expected to satisfy two fundamental properties: *Completeness* and *Transitivity*. As all decision steps are estimated their utilities as Elo scores numerically, these two properties are satisfied naturally for the agent. For *Completeness*, any two decisions can be compared numerically based on their Elo scores to determine their relative superiority. For *Transitivity*, it is obvious that given three decision steps $A, B, C$, if $v_A > v_B$ and $v_B > v_C$, the Elo score of $A$ must be greater than $C$ ($v_A > v_B > v_C$). Consequently, the agent has constructed the rationality that allows it to select the best-performed one as the final solution. Specifically, given all existing decision sequences $\mathbb{T} = \{t_1, t_2, \cdots, t_n\}$, the one which final decision with the highest utility is selected as the final solution.

$$t = \arg\max_{t \in \mathbb{T}} \{V(d_N)\} \tag{9}$$

where $d_N$ refers to the final decision step.

## 4 EXPERIMENT

As the key contribution of this work is to develop a rational decision-making agent with internalized utility judgment, we aim to answer the following research questions through a series of experiments.

**RQ1** Can RADAGENT make decisions rationally to accomplish a diverse set of tasks?

**RQ2** Beyond finding feasible solutions, can RADAGENT find better solution?

**RQ3** How efficient is RADAGENT in decision making?

**RQ4** Is Elo-based Utility Construction effective in providing reliable utility assessments?

**RQ5** What are the key differentiating factors of RADAGENT against other methods?

Next, we describe the experimental settings and then report results by answering these questions.

### 4.1 EXPERIMENTAL SETTINGS

**Datasets** We conduct extensive experiments on Game of 24 (Yao et al., 2023), WebShop (Yao et al., 2022a), and ToolBench (Qin et al., 2023c) datasets. Game of 24 aims to use 4 numbers and four fundamental arithmetic operations ($+ - */$) to reach 24. WebShop focuses on simulating the process of searching, browsing, and selecting items on an online shopping platform in order to obtain a desired item. For ToolBench, we focused on the intra-category multi-tool instruction scenario which has thoughtfully constructed a diverse and intricate collection of human instructions of over 16K APIs from 49 categories. It accurately reflects the complexities involved in real-world tasks, necessitating the use of various tools and requiring multi-step decision-making processes. We use 100, 500, 500 instances for Game of 24, WebShop, and ToolBench to evaluate the decision-making ability respectively.

| Model | Game of 24 | WebShop | ToolBench |
|---|---|---|---|
| CoT | 6.00 | 56.23 | 16.60 |
| CoT@3 | 7.00 | 56.45 | 31.20 |
| Reflexion | 7.00 | 57.21 | 26.60 |
| BFS | 11.00 | 50.20 | 38.00 |
| DFS | 14.00 | 55.60 | 45.58 |
| DFSDT | 29.00 | 57.25 | 50.20 |
| RADAGENT | **43.00** | **59.36** | **61.92** |

Table 1: Main experimental results on Game of 24, WebShop, and ToolBench dataset. Bold marks the best performance.

| Model | Pref. Rank |
|---|---|
| CoT@3 | 3.45 |
| Reflexion | 3.48 |
| BFS | 3.25 |
| DFSDT | 2.91 |
| RADAGENT | |
| *-RandSelect* | 3.24 |
| *-EloSelect* | **2.19** |

Table 2: Solution ranking experimental results on ToolBench dataset. Bold marks the top rank.

**Baselines**   We compare RADAGENT with the following decision-making methods: (1) **CoT** (Wei et al., 2023; Yao et al., 2022b) decomposes reasoning into explicit intermediate steps and we adapt ReACT (Yao et al., 2022b) to make sequential decisions. (2) **CoT@3** extends the CoT approach by running the decision-making process three times independently for an instruction and finally generates a total of three decision sequences. (3) **Reflexion** (Shinn et al., 2023) builds upon CoT@3 and allows LLMs to engage in self-reflection on their previous decision sequences. (4) **BFS** (Yao et al., 2023) constructs a decision tree in a top-down manner to search for a feasible solution. (5) **DFS** (Yao et al., 2023) constructs a decision tree by going as deep as possible along each branch and exploring the most recently visited states. (6) **DFSDT** (Qin et al., 2023c) is an improved version of DFS, which allows agents to dynamically assess different decision states and choose to either proceed along a promising path or abandon an existing state and expand another one.

**Evaluation Metrics**   To ensure a rigorous and accurate evaluation of the performance of our proposed decision-making approach, we adopt three evaluation metrics for each dataset respectively: (1) **Success Rate** (Yao et al., 2023) measures the proportion of valid equations generated by the agent's arithmetic operations that yield a result of 24, using the given input numbers. (2) **Reward** (Yao et al., 2022a) evaluates the similarity (a value between 0 and 1) between the attributes of the items chosen by the agent and the attributes of the items actually purchased by human. (3) **Pass Rate** (Qin et al., 2023c) assesses the ability of agents to successfully accomplish complex real-world tasks by using tools sequentially. It calculates the proportion of instructions that an agent completed.

**Implementation Details**   We use ChatGPT (`gpt-3.5-turbo-0613-16k`) to implement our approach. Our approach involves conducting a decision-exploration process 20 times and finally selecting the decision sequence with the highest Elo score as the final decision. For Elo-based Utility Construction, the initial Elo score of the decision step is set as $0.0$ and the Elo coefficient $r$ is set as $173.72$ according to the vanilla Elo rating system (Elo, 1967). The Elo score of $\hat{d}$ in Equation 5 is set as $0.0$. $K$ in Equation 3 is set as $50$. To manage the computational cost of ChatGPT API calls, we set a limit of 100 ChatGPT API calls for a decision-searching process. Furthermore, we impose a maximum limit of 12 steps for each decision sequence due to the cost of ChatGPT API calls.

## 4.2   OVERALL RESULTS (RQ1)

To validate the effectiveness of our proposed RADAGENT approach, we first study whether our approach can accomplish more complex tasks. The results are shown in Table 1 and we can observe that: (1) Across all datasets, RADAGENT consistently outperforms all baselines, indicating that incorporating the utility judgment internally empowers agents to accomplish a broader range of tasks effectively. (2) In Game of 24 and ToolBench domains, RADAGENT exhibits the capability to assign lower elo scores to decision steps that lead to failure. Consequently, these Elo scores serve as guidance for agents to avoid such decisions and achieve success. (3) For Webshop, while our method still outperforms all baselines, it only achieves only marginal gains. This is attributed to the fact that Webshop provides only one "golden answer" for each instruction, while several other items actually meet the requirements. Consequently, these alternative items receive lower rewards as they deviate from the golden answer, resulting in an underestimation of the performance.

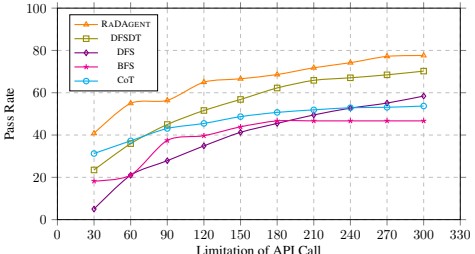

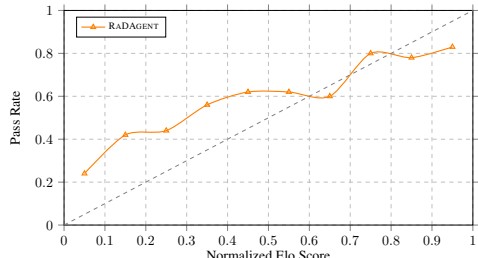

Figure 2: Efficiency experimental results on various API cal limitations.

Figure 3: Performance on different data split with varied Elo scores.

### 4.3 SOLUTION RANKING (RQ2)

In addition to validating the effectiveness of our approach to reach feasible solutions, we seek to investigate whether RADAGENT can further provide solutions with higher quality. We adopt *Win Rate* in ToolEval from ToolBench to compare the decision sequences produced by different methods for a given instruction. Based *Win Rate*, we utilize PRP (Qin et al., 2023d) to rank decision sequences of all methods to report their rank to measure the superiority of each decision sequence. We further develop a variant of our model named *RandSelect* which selects the final decision sequence randomly while *EloSelect* selects based on the highest Elo score. We then select representative baselines (CoT@3, Reflexion, BFS, DFS, DFSDT) and conduct a comprehensive comparison of the decision sequences produced by each method. The experimental results are summarized in Table 2, and it reveals that RADAGENT consistently achieves the top average rank among all comparable baselines. Especially, *EloSelect* obviously outperforms *RandSelect*, confirming the capability of our Elo-based Utility Construction to assess the utility of each decision sequence to select superior solutions, resulting in high-quality decision making.

### 4.4 EFFICIENCY ANALYSIS (RQ3)

We further conducted the analyses to evaluate the efficiency of our proposed RADAGENT. As making a decision step will involve a ChatGPT API call, the inefficient decision-making method would involve more API calls to accomplish the same instruction, causing costly expenses. We thus conducted experiments with varying ChatGPT API call limitations, ranging from 30 to 300, and measured Pass Rate in ToolBench of each method under these varied limitations. The experimental results are demonstrated in Figure 2. These results showcase that the BFS, DFS, and DFSDT heavily rely on a large number of ChatGPT API call to achieve a high Pass Rate. Once limiting the number of API calls, their performance even cannot surpass CoT. In contrast, our approach achieves the highest Pass Rate under all limitation settings, especially in low-resource settings. We attribute it to the fact that our method can utilize Elo scores to dynamically select the promising decision steps, avoiding those unpromising ones. Thus, our method illustrates superior efficiency against baselines and the practical advantages of our approach in real-world scenarios.

### 4.5 RELIABLE UTILITY ASSESSMENT OF ELO SCORE (RQ4)

To verify the effectiveness of our Elo-based Utility Construction in providing reliable utility assessments, we conducted a comprehensive analysis using the ToolBench dataset. As the Elo score serves as a metric to represent the utility of each decision, we seek to determine whether the Elo score is a reliable indicator of decision utility. To this end, we partitioned the ToolBench dataset into several subsets based on the Elo scores assigned to the decision sequences generated by RADAGENT. We first collect the Elo scores for all ToolBench data and then normalized them to scale within the range of 0 to 1. Next, we sort the normalized Elo scores and divided them into 10 intervals, getting 10 subsets of ToolBench data accordingly and calculated the Pass Rate for each method on these 10 subsets. Figure 3 illustrates the experimental results. A discernible trend is observed across all methods: the Pass Rate consistently increases with higher Elo scores. This clear positive correlation between the Elo score and the Pass Rate demonstrates the efficacy of the Elo-based Utility Con-

| Method | Hallucinated Tool | | Tool Call Error | | Unavailable Tool | Decision Failure |
|--------|-----------|------|-----------|------|------------------|------------------|
| | Incidence | Fix | Incidence | Fix | | |
| CoT@3 | 14.2 | 25.4 | 41.2 | 14.8 | 2.0 | 52.5 |
| BFS | 18.8 | 25.5 | 50.8 | 31.1 | 2.6 | 48.6 |
| DFSDT | 31.5 | 38.9 | 62.5 | 41.0 | 3.0 | 26.4 |
| RADAGENT | 42.1 | **53.3** | 62.3 | **54.0** | 3.0 | **14.8** |

Table 3: Incidence ratio and Fix ratio of common failure reasons in ToolBench dataset.

struction in providing reliable assessments of decision utility. A higher Elo score indicates that the decision sequence is more likely to represent an accomplished solution to the instruction, whereas a lower Elo score suggests that the instruction may be more challenging, and the corresponding decision sequence may not effectively solve the instruction.

### 4.6 ERROR ANALYSIS (RQ5)

In this section, we present a comprehensive case analysis to elucidate the specific tasks that RADA-GENT effectively addresses. By dissecting the nature of RADAGENT's successes and failures, we shed light on its decision-making capabilities and limitations.

We commence our analysis by categorizing the common reasons for failure encountered by each model in ToolBench. These reasons encompass: (1) **Unavailable Tool**: Occurrences where a subset of the designated tools is inaccessible, e.g., HTTP 404 or 500 error. (2) **Tool Call Error**: Instances of tool call errors, including issues related to parameter format mismatching and missing mandatory parameter fields. (3) **Hallucinated Tool**: Instances where the model employs tools not provided, i.e., invoking a non-existent tool. (4) **Decision Failure**: Instances where the model fails to accomplish although none of the aforementioned problems occur. We present the incidence ratio of the aforementioned categories. As Hallucinated Tool and Tool Call Error can be fixed to some extent by agents during decision making, we further report the fix ratio that models successfully fix the occurred errors to accomplish the instructions to represent the ability to handle exceptions.

From Table 3, several noteworthy observations arise: (1) RADAGENT boasts the lowest incidence ratio of decision failure, highlighting its adeptness in decision making. (2) RADAGENT outperforms other methods significantly in fixing tool call errors and hallucinated tools, demonstrating the robustness of our method as it can learn a lower utility for these failure to guide the decision making to avoid them. (3) All methods own similar incident ratio of unavailable tool which shows that there still exist some inoperative APIs in ToolBench, influencing the decision-making process. (4) We examine cases that all methods fail and find certain cases remain unsolvable due to the ambiguity of user-provided values (e.g., user ID, email address) or restrictions imposed by limited tool chain lengths, which underscores the necessity for advanced decision-making proficiencies.

### 4.7 DISCUSSION

In this section, we discuss the multifaceted competencies that a decision-making agent necessitates.

- Exception Handling. During the decision-making process, exceptions may occur (e.g., tool unavailable, tool call errors), leading to the decision step not meeting the expectation. Under these circumstances, decision-making methods should have the ability to deal with the exceptions to navigate to a new decision. CoT is susceptible to these scenarios, which leads the model into a loop of repeated erroneous decisions. In contrast, BFS, DFS, DFSDT, and our method excel in mitigating such occurrences as they can explore potential decisions to avoid exceptions.
- Diversity Exploration. There exist different decision directions to accomplish tasks. For example, in tool use scenarios, some tools have analogous functionalities and one of them is the most functional to accomplish tasks. DFS and DFSDT, constrained by their relatively narrow search width, might miss identifying the optimal decision. Although BFS can make several decisions in a step, it fails to explore promising decisions efficiently. In contrast, RADAGENT assigns lower scores to fewer potential decision steps, displaying a trend for exploring novel avenues, which exemplifies a scenario demanding diversity in exploration.

- Decision Reflection. Complex tasks should be divided into sequential decisions and the model should accomplish them progressively to finish the task finally. It requires models to verify the completeness of each decision step and reflect to make better decisions toward successful directions accordingly. DFSDT cannot evaluate the intermediate decision so it cannot learn a good reflection from previous decisions to select an effective one. RADAGENT, benefitting from its internalized utility judgment, assigns higher scores to decision steps aligned with comprehensive solution strategies. This innate ability to recognize the completeness of previous decisions and guide the next decision accordingly is a hallmark of an effective decision-making method.

## 5 RELATED WORK

**Decision-Making Methods for LLM-based Agents**   Efficient and effective decision-making ability is fundamental for LLM-based agents to the attainment of specific objectives (Yao et al., 2022b; 2023; Hao et al., 2023a; Besta et al., 2023; Sel et al., 2023). Although LLMs are pre-trained on a large-scale corpus which equips them with substantial common sense and knowledge to solve several problems, due to the complexity and diversity of realistic tasks, LLM-based agents still struggle to make multi-step decisions to solve realistic tasks. Recently, as Chain-of-Thought (Wei et al., 2023) demonstrates its capability to decompose complex questions into sequential intermediate steps, several LLM-based decision-making methods are proposed to enhance the decision-making ability of agents. ReACT (Yao et al., 2022b) develops a variant of CoT to leverage the reasoning ability of LLMs in decision-making scenarios. Reflexion (Shinn et al., 2023) further offers a remedial approach to make LLMs reflect their failure and summarize the reason in the decision process, and then correct their mistake in the second attempt. Based on these methods, some tree-based decision-making methods are proposed to adapt the decision-making ability of LLMs into specific tasks. Tree-of-Thought (Yao et al., 2023) proposes BFS and DFS decision-making algorithms in Game of 24, Creative Writing and Mini Crosswords tasks. RAP (Hao et al., 2023a) applies the Monte Carlo Tree search algorithm to find a good solution in Blocksworld, Math Reasoning, and Logical Reasoning tasks. DFSDT (Qin et al., 2023c), following a similar tree search algorithm, proposes an efficient version of DFS to make decisions. However, the aforementioned methods need external performance measure to guide the decision-making process, which limits their scope of application. In this paper, we propose RADAGENT which internalizes the utility judgment ability with Elo rating system to achieve rationality for agents to provide optimal solutions.

**Tool Learning**   Recent investigations have cast illumination upon the burgeoning proficiencies exhibited by LLM-based agents in the mastery of instruments and the execution of decision-making processes within intricate contextual milieus (Qin et al., 2023b; Vemprala et al., 2023; Nakano et al., 2021; Qin et al., 2023a; Shen et al., 2023; Wu et al., 2023; Schick et al., 2023; Hao et al., 2023b; Qian et al., 2023; Song et al., 2023; Qin et al., 2023c). The incorporation of external tools into the operational framework of LLM-based agents confers upon them immediate access to contemporaneous factual knowledge (Yang et al., 2023), imbues them with versatile multimodal capabilities (Gupta & Kembhavi, 2023), and empowers them with specialized proficiencies tailored to vertical domains (Jin et al., 2023). However, when confronted with real-world tasks that often require the utilization of multiple tools, LLM-based agents must engage in multi-step decision-making processes to select tools and determine their sequencing. Consequently, the ability for decision-making in tool learning scenarios becomes imperative to effectively tackle practical applications.

## 6 CONCLUSION

In this work, we have introduced a novel approach, RADAGENT, to internalize the utility judgment ability for agents to achieve rationality across a diverse range of real-world tasks. The introduction of an Elo-based Utility Construction enhances agents to learn numeric utility for each decision step and guide the decision-making process. Extensive experiments on the Game of 24, WebShop, and ToolBench dataset have confirmed the effectiveness of RADAGENT, outperforming baseline methods by achieving notable improvements and producing higher-quality solutions. Moreover, the reduction in LLM API calls showcases the efficiency gains of our approach. By empowering agents with rationality, our work paves the way for their broader utilization in real-world scenarios, alleviating the reliance on external performance measure.

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

## A  UTILITY LEARNING PROMPT

Our utility learning prompt is designed as follows:

```
You are value-GPT, an expert in defining which trail is better and
    closer to solving the task. Here is the task description:
*******************************
{{BEGIN_DESCRIPTION}}
your_task: {task_description}
your_query: {input_description}
{{END_DESCRIPTION}}
*******************************
Here are two candidates A and B. They both try to handle the task
    with some function calls. Their trails are as follows.
*******************************
{{CANDIDATE_A_START}}
{candidate_A}
{{CANDIDATE_A_END}}
*******************************
{{CANDIDATE_B_START}}
{candidate_B}
{{CANDIDATE_B_END}}
*******************************
```

Then, ChatGPT should call the following function[1] to give the result.

```
{
    "name": "choose_preference",
    "description": "Choose the preferred answer for the query
        within all given answers.",
    "parameters": {
        "type": "object",
        "properties": {
            "preference": {
                "type": "number",
                "description": "The index of the preferred answer
                    in all given answers."
            },
        },
    },
}
```

---

[1]https://openai.com/blog/function-calling-and-other-api-updates

