# OpenReview forum: "Rational Decision-Making Agent with Internalized Utility Judgment"
_ICLR.cc/2024/Conference — Submitted to ICLR 2024_

### Official Review · Reviewer_4ygb · 2023-10-22

**Soundness:** 3 good
**Presentation:** 2 fair
**Contribution:** 3 good
**Rating:** 6
**Confidence:** 3

**Summary:**

This paper proposes RADAGENT, a Large-Language-Model(LLM)-based decision-making agent that does not need external performance feedback. RADAGENT explores in a tree stucture similar to Tree-of-Thought and learns a utility of solutions based on ELO scores over different outcomes. The ELO score is calculated by LLM-prompted pairwise comparison between solutions, and guides the annealed probability of choosing branches for exploration. The result with the best ELO score is used as the final solution. On sevaral decision-making environments, RADAGENT outperforms ReAct, Tree-of-Thought-based methods and Reflexion.

**Strengths:**

1. The problem studied by this paper, how to improve the decision-making ability of Large Language Model (LLMs) with no external feedback, is timely and interesting. For example, on some reasoning environments that can be modeled as decision-making ones such as HotPotQA [1], it is very reasonable that the agent does not know whether its answer is correct until it gives the final answer. External feedback would also make tasks such as fact verification [2] and multiple choices [3] trivial, and thus decision-making with no external feedback is important.

2. The proposed solution based on ELO scores and tree-of-thought is simple, intuitive, yet effective on multiple environments.

**References**

[1] Z. Yang et al. Hotpotqa: A dataset for diverse, explainable multi-hop question answering. arXiv:1809.09600, 2018.

[2] J. Throne et al. FEVER: a Large-scale Dataset for Fact Extraction and VERification. In NAACL, 2018.

[3] T. Yahmed. MCQA. https://github.com/mcQA-suite/mcQA, 2019.

**Weaknesses:**

**1. The presentation of the work is not clear enough.**

a) The state, action and decision step in the context is not well-defined. As Large Language Models (LLMs) take text as input and output text, it is unclear what do state and action correspond to in the context. I assume the authors develop their method based on ReAct [1] and Tree-of-Thought [2] as illustrated in Figure 1, but they are not properly introduced in the methodology part. In addition, I would recommend the authors to emphasize that the reward is not given in Markov Decision Process (MDP) of this paper, which is uncommon.

b) The "subsequent step" in Sec. 3.1 is unclear. In preliminary section, decision steps have a subscript that stands for the number of steps, and "subsequent step" can be comprehended as a chain of steps following the current step. but in Sec. 3.1, the subsequent step with different subscript should be different versions of next step. Also, I would suggest to introduce $n$, change $\hat{d}$ to $d_{n+1}$ (because $\hat{d}$ has no subscript), and change $\sum_j$ to $\sum_{j=1}^{n+1}$.

**2. Prompts for each environment is not specified.** Only the prompt for utility learning is listed; the prompt for each environment is missing in the appendix.

**3. The authors could perform more ablations on the exploration process.** The exploration process is controlled by multiple factors, such as temperature $\tau$, initial ELO score $\hat{v}$ and ELO coefficient $r$, and because the probability of exploring a new decision is exponential to the initial ELO score, it is intuitive that RADAGENT might fall into local minima where the currently best solution presents in the branch but can still be improved. Thus, it would be better if an ablation study on the hyperparameter could be performed to better measure the robustness of RADAGENT.

**References**

[1] S. Yao et al. React: Synergizing reasoning and acting in language models. ArXiv preprint, abs/2210.03629, 2022.

[2] S. Yao et al. Tree of thoughts: Deliberate problem solving with large language models. arXiv preprint arXiv:2305.10601, 2023.

**Questions:**

I have one question: how does the method work with GPT-4, which I expect would give us even better performance? Also, I would suggest the authors to include a pseudocode in the appendix to help the readers better understand the algortihm process.

---

> ### Author Response · Authors · 2023-11-22
>
> Q1: The presentation of the work is not clear enough.
>
> A1: Thanks for your suggestions about providing adequate background information to contextualize our methodology. To this end, we plan to reorganize the structure of our paper, placing the Related Work section after the Introduction in the final version. This will offer readers a more comprehensive understanding of existing approaches like ReAct and Tree-of-Thought, thereby facilitating a clearer grasp of the decision-making process in our RaDAgent framework.
>
> For the fundamental concepts of MDP (i.e., state, action, reward), we would like to elucidate them. These concepts have different meanings in different scenarios.
> - In the Game of 24 scenario:
>     - _State_ refers to the current calculation result of the four numbers.
>     - _Action_ denotes the selection of one of four arithmetic operations (+, −, ∗, /).
> - In the Webshop scenario:
> 	- _State_ refers to the current web page information.
> 	- _Action_ refers to the web browsing operations (e.g., click, go back, purchase)
> - In the ToolBench scenario:
>     - _State_ encompasses the context of the current decision sequence, including historical API calls and their corresponding results.
>     - _Action_ refers to the APIs that RaDAgent can call.
>
> The concept of _Reward_ in our work diverges from the **traditional external performance measures** typical in common MDP approaches. Instead, we focus on showcasing the importance of **internal rationality** (i.e., internalized utility judgment). In this light, the Elo scores in RaDAgent serve as a form of 'reward', reflecting the utility judgment capability of the agent. These scores are internally derived, aligning with our goal of demonstrating self-sufficient decision-making prowess without reliance on external validations. We will incorporate these clarifications in the revised manuscript, ensuring that the definitions of state, action, and reward are explicitly tailored to their specific applications in the various scenarios we explore.
>
> For the notations about "subsequent step" in Section 3.1, we are grateful for your observation and suggestions. We will revise the manuscript to refine the symbols and terms used to describe the decision steps.

---

> ### Author Response · Authors · 2023-11-22
>
> Q2: Prompts for each environment is not specified.
>
> A2: Apologize for the absence of prompts. We list all prompts of our method and experiments as follows.
>
> The decision-making prompt which is used to make decisions during the experience exploration process is listed as follows:
> ```
> You are the Decision-Making GPT and can perform any task using the tree search method.
> The search method is as follows:
> 1. First, I will provide you with the task description and input details.
> 2. For each task, you need to call various functions through multiple steps.
> 3. At each step, you need to give your thought to analyze the status now and what to do next, with a function call to actually execute your step.
> After the call, you will get the call result, and you are now in a new state.
> 4. Each (thought-function) pair mentioned above is considered a tree node, and each trail is a tree path from the root to a terminal node. Therefore, the Monte Carlo search tree contains multiple trails.
> 5. Although you may not see previous trails, in each trail, I will first place you in an intermediate state determined by the "value" of the node, and then you make different choices from there.
>
> Remember:
> 1. Always make a function call at each step.
> 2. If you believe you have gathered enough information, call the function "Finish: give_answer" to provide your answer for the task.
> 3. If you feel unable to handle the task from this step, call the function "Finish: give_up_and_restart".
>
> Let\'s begin!
> Task description: {task_description}
> ```
>
> The task description of Game of 24 is as follows:
> ```
> Use numbers and basic arithmetic operations (+ - * /) to obtain exactly one number 24. In each step, you are only allowed to choose two of the left numbers to obtain a new number. For example, 7 * 9 - 3 * 1 3 = 2 4.
> Remember:
> 1. All of the numbers must be used and must be used ONCE. So Only when the left number is exactly 24, you will win. So you don't succeed when the left number = [2 4, 5]. You succeed when the left number = [2 4].
> 2. All the try takes exactly 3 steps, and the task ends when the count of left numbers is 1, and check whether the only left number equals 24.
> 3. When there are two numbers left, ALWAYS pre-compute and list all the operations' combined results( + - * / ), and find if there is a way to combine them to 24 before you make the function call.
> 3.1. If There is a way, use function "play_24" to combine it.
> 3.2. If not, use function "give_up" to restart.
> 4. The status changes ONLY when you call function "play_24". If you only give thoughts, nothing happens.
> 5. "play_24" inputs only one step, if you want to combine many numbers, split it into multiple calls.
> ```
>
> The task description of Webshop is listed as follows:
> ```
> You should use functions to help handle the web shopping task on a webshop site.
> We have 2 Pages: Product Selection Page & Product Details Page.
> You have access to the following functions:
> 1. search: at any time, you can search a product by keywords. Then you will go to the Product Selection Page which shows a list of related products.
> 2. select: after searching keywords, you can select a product on the Product Selection Page. Then you will go to the Product Details Page, which shows the details of the product you select.
> 3. buy: On the Product Details Page, you can buy a product. Then the shopping task is completed.
> ```
>
> The task description of ToolBench is listed as follows:
> ```
> You should use functions to help handle real-time user queries. Remember:
> 1. ALWAYS call \"Finish\" function at the end of the task. And the final answer should contain enough information to show to the user, If you can't handle the task, or you find that function calls always fail(the function is not valid now), use function Finish->give_up_and_restart.
> 2. Do not use origin tool names, use only subfunctions' names.
> You have access to the following tools:\n
> ```
>
> These prompts will be added in the Appendix section in the final version to provide more experimental details. Moreover, we plan to release all code including the method implementation and experiment settings on Github to ensure reproducibility after acceptance.

---

> ### Author Response · Authors · 2023-11-22
>
> Q3: The authors could perform more ablations on the exploration process.
>
> A3: For your questions about the robustness of RaDAgent in terms of hypermeters (including temperature $\tau$, initial Elo score $\hat{v}$, and Elo coefficient $r$), I want to make clarification.
>
> First, the selection of $r$ is based on the classic Elo rating algorithm. In the classic Elo rating system[1,2,3], the expected superiority is defined as:
> $$
> E = \frac{1}{1+10^{-\frac{(R1-R2)}{400}}}
> $$
> In this paper, we employ the base $e$ to compute the expected superiority for computational implementation convenience:
> $$
> E = \frac{1}{1+e^{-\frac{(R1-R2)}{K}}}
> $$
> To achieve equivalence between them, we can set $K$ as approximately $K = \frac{400}{\ln10} \approx 173.72$ to change the base. In this way, our calculation of the expected superiority equals the classic Elo rating algorithm.
>
> [1] Elo et al., The rating of chessplayers: Past and present. 1978.
>
> [2] Elo et. al., Logistic probability as a rating basis. 1986.
>
> [3] The US Chess Rating system. http://www.glicko.net/ratings/rating.system.pdf, 2017.
>
> Second, for the temperature $\tau$, it is initialized as $\tau = 173.72$ which is aligned with Elo coefficient $r$. This choice is motivated by our intention to base the exploration process on the expected superiority of decision steps. After multiple Elo comparisons have conducted, the Elo scores gradually converge to the accurate value. We draw inspiration from the Monte Carlo Tree Search algorithm[4,5] to anneal the exploration probability, thereby directing the exploration process towards more favorable decisions.
>
> [4] Coulom et al., Efficient Selectivity and Backup Operators in Monte-Carlo Tree Search. Computers and Games. 2006.
>
> [5] Silver et al., Mastering the game of Go with deep neural networks and tree search. Nature. 2016.
>
> Third, for the initial Elo score of $\hat{d}$, we want to clarify the role of the initial Elo score in our RaDAgent. It is crucial to understand that in the Elo rating algorithm, the decision-making process is influenced by the relative difference between Elo scores of decision steps, rather than their absolute values. You can refer to Equation 1 in our paper. Furthermore, the essence of the Elo rating system lies in its dynamic nature, where scores are adjusted based on pairwise comparisons over time. This means that regardless of the initial score assigned to each decision step, the subsequent adjustments made through pairwise comparisons are what determine the final, accurate assessment of each decision's utility. Therefore, the initial score primarily serves as a starting point, and its specific value is not critical to the overall decision-making process. Consequently, the initial Elo score assigned to decision steps does not fundamentally impact the outcome of the comparisons.
>
> To further assess the impact of these hyperparameters, we plan to conduct additional experiments in the future. We appreciate your valuable suggestions and will incorporate them into our future work.

---

> ### Author Response · Authors · 2023-11-23
>
> Q4: How does the method work with GPT-4, which I expect would give us even better performance?
>
> A4: We have conducted additional experiments integrating GPT-4 into our RaDAgent on the Game of 24 scenario . The experimental results are shown in the following Table:
>
> | Method  | Succeed Rate |
> |---------|-----------|
> | ChatGPT | 43.0      |
> | GPT-4   | 54.0      |
>
> We found that RaDAgent, leveraging the enhanced capabilities of GPT-4, demonstrates its superiority over its ChatGPT version. This finding underscores the scalability and adaptability of RaDAgent. It suggests that as LLMs continue to become more sophisticated, our model can leverage these advancements to achieve even greater levels of decision-making efficiency and accuracy.

---

> > ### Comment · Reviewer_4ygb · 2023-11-23
> > **Response to Authors**
> >
> > Thanks for the authors' detailed response, which addresses my concern. I decide to keep the current score.

---

### Official Review · Reviewer_1oKH · 2023-10-30

**Soundness:** 3 good
**Presentation:** 4 excellent
**Contribution:** 3 good
**Rating:** 6
**Confidence:** 4

**Summary:**

The authors introduce a novel rational decision-making agent, RaDAgent, which seeks to instill rationality in large language models (LLMs) through internalized utility judgment. RaDAgent operates on an iterative framework that encompasses experience exploration and utility learning to establish this internalized utility judgment. During experience exploration, the agent samples the next decision step from a Boltzmann distribution, which is determined by their Elo scores. To learn the Elo score function, the method employs pairwise comparison and subsequent score update. It starts by recalibrating the Elo scores of the final decision steps in each sequence based on pairwise comparison, then updates the Elo scores of preceding steps using the scores of their subsequent decision steps. Various experimental tasks validate RaDAgent's superiority over its competitors.

**Strengths:**

1. The paper is articulate, presenting high-quality content that is easy to follow.
2. The methodology proposed is novel. By leveraging LLM’s inherent capability for value assessment, it pioneers a way to guide decision-making without the need for manually tailored prompts for value evaluation.
3. The experimental results robustly corroborate the efficacy of the proposed method. RaDAgent consistently surpasses the benchmarks in various tasks. Moreover, the correlation observed between Elo scores and pass rates further underlines the effectiveness of the Elo-based Utility Construction in gauging decision utility.

**Weaknesses:**

1. The revelation of the experimental details is inadequate. Notably, not all prompts used are revealed, and no examples are provided. Such omissions pose a challenge for reproducibility and make it difficult to pinpoint the source of the observed improvements.
2. There's a conspicuous absence of an ablation study concerning the Elo-based value evaluation. By not contrasting it with a manual value evaluation prompt, it remains ambiguous whether the observed performance boost arises from the Elo-based valuation, the experience exploration, or both.

**Questions:**

1. I suggest revealing all experimental details so that the contribution can be properly evaluated and reproduced.
2. It would be helpful to conduct an extra experiment comparing the Elo-based evaluation and manually designed value evaluation.
3. Why is the Boltzmann distribution used? How about using upper confidence tree for searching?

---

> ### Author Response · Authors · 2023-11-22
>
> Q1: The experimental details if inadequate. Notably, not all prompts used are revealed, and no examples are provided.
>
> A1: Apologize for the inadequate experimental details. We list all prompts of our method and experiments as follows.
>
> The decision-making prompt which is used to make decisions during the experience exploration process is listed as follows:
> ```
> You are the Decision-Making GPT and can perform any task using the tree search method.
> The search method is as follows:
> 1. First, I will provide you with the task description and input details.
> 2. For each task, you need to call various functions through multiple steps.
> 3. At each step, you need to give your thought to analyze the status now and what to do next, with a function call to actually execute your step.
> After the call, you will get the call result, and you are now in a new state.
> 4. Each (thought-function) pair mentioned above is considered a tree node, and each trail is a tree path from the root to a terminal node. Therefore, the Monte Carlo search tree contains multiple trails.
> 5. Although you may not see previous trails, in each trail, I will first place you in an intermediate state determined by the "value" of the node, and then you make different choices from there.
>
> Remember:
> 1. Always make a function call at each step.
> 2. If you believe you have gathered enough information, call the function "Finish: give_answer" to provide your answer for the task.
> 3. If you feel unable to handle the task from this step, call the function "Finish: give_up_and_restart".
>
> Let\'s begin!
> Task description: {task_description}
> ```
>
> The task description of Game of 24 is as follows:
> ```
> Use numbers and basic arithmetic operations (+ - * /) to obtain exactly one number 24. In each step, you are only allowed to choose two of the left numbers to obtain a new number. For example, 7 * 9 - 3 * 1 3 = 2 4.
> Remember:
> 1. All of the numbers must be used and must be used ONCE. So Only when the left number is exactly 24, you will win. So you don't succeed when the left number = [2 4, 5]. You succeed when the left number = [2 4].
> 2. All the try takes exactly 3 steps, and the task ends when the count of left numbers is 1, and check whether the only left number equals 24.
> 3. When there are two numbers left, ALWAYS pre-compute and list all the operations' combined results( + - * / ), and find if there is a way to combine them to 24 before you make the function call.
> 3.1. If There is a way, use function "play_24" to combine it.
> 3.2. If not, use function "give_up" to restart.
> 4. The status changes ONLY when you call function "play_24". If you only give thoughts, nothing happens.
> 5. "play_24" inputs only one step, if you want to combine many numbers, split it into multiple calls.
> ```
>
> The task description of Webshop is listed as follows:
> ```
> You should use functions to help handle the web shopping task on a webshop site.
> We have 2 Pages: Product Selection Page & Product Details Page.
> You have access to the following functions:
> 1. search: at any time, you can search a product by keywords. Then you will go to the Product Selection Page which shows a list of related products.
> 2. select: after searching keywords, you can select a product on the Product Selection Page. Then you will go to the Product Details Page, which shows the details of the product you select.
> 3. buy: On the Product Details Page, you can buy a product. Then the shopping task is completed.
> ```
>
> The task description of ToolBench is listed as follows:
> ```
> You should use functions to help handle real-time user queries. Remember:
> 1. ALWAYS call \"Finish\" function at the end of the task. And the final answer should contain enough information to show to the user, If you can't handle the task, or you find that function calls always fail(the function is not valid now), use function Finish->give_up_and_restart.
> 2. Do not use origin tool names, use only subfunctions' names.
> You have access to the following tools:\n
> ```
>
> These prompts will be added in the Appendix section in the final version to provide more experimental details. Moreover, we plan to release all code including the method implementation and experiment settings on Github to ensure reproducibility after acceptance.

---

> ### Author Response · Authors · 2023-11-22
>
> Q2: How about using upper confidence tree for searching?
>
> A2: We are thankful for your suggestion to consider the upper confidence tree (UCT) approach. UCT is indeed a powerful mechanism that uses multiple Monte Carlo simulations to accurately assess the utility of each decision node, with the precision of assessment improving as the number of simulations increases. However, in the context of LLM-based decision-making, conducting simulations for each decision step would entail numerous ChatGPT API calls. This would not only prove inefficient but also costly. Instead, we utilize LLM comparisons to estimate the utility of decision steps. By leveraging the Elo algorithm, which inherently provides a measure of expected performance based on historical comparisons, we achieve significant efficiency. The Elo algorithm's capacity for rapid utility evaluation without the need for extensive simulations is particularly advantageous in our application. Our experiments in Section 4.4 also demonstrate the superiority of RaDAgent in terms of efficiency.

---

> ### Author Response · Authors · 2023-11-22
>
> Q3: Why is the Boltzmann distribution used?
>
> A3: The Boltzmann distribution's application in RaDAgent directly stems from the foundational principles of the Elo rating algorithm. Traditionally, the Elo algorithm calculates the expected superiority using the base 10[1,2,3].
> $$
> E_{R_1 > R_2} = \frac{1}{1+10^{-\frac{(R_1-R_2)}{400}}} = \frac{10^{-\frac{R_2}{400}}}{10^{-\frac{R_1}{400}} + 10^{-\frac{R_2}{400}}}
> $$
> However, for computational expediency and implementation simplicity in our system, we adjusted the base from 10 to the natural constant $e$:
> $$
> E_{R_1 > R_2} = \frac{1}{1+e^{-\frac{(R_1-R_2)}{173.72}}} = \frac{e^{-\frac{R_2}{173.72}}}{e^{-\frac{R_1}{173.72}} + e^{-\frac{R_2}{173.72}}}
> $$
> This modification naturally led to an expression resembling the Boltzmann distribution when calculating expected superiority probabilities.
>
> The resemblance between the Boltzmann distribution and our adapted Elo calculation is serendipitous yet profoundly interesting, bridging concepts from statistical mechanics and rational decision-making. Your insightful observation has highlighted a promising research direction. We intend to delve deeper into this connection and analyze the Boltzmann distribution's implications in the context of the Elo rating algorithm. Such an exploration may indeed reveal intrinsic properties of the Elo algorithm that are not only theoretically intriguing but could also enhance its practical application in LLM-based decision-making agents.
>
> [1] Elo et al., The rating of chessplayers: Past and present. 1978.
>
> [2] Elo et. al., Logistic probability as a rating basis. 1986.
>
> [3] The US Chess Rating system. http://www.glicko.net/ratings/rating.system.pdf, 2017.

---

> ### Author Response · Authors · 2023-11-22
>
> Q4: There's a conspicuous absence of an ablation study concerning the Elo-based value evaluation.
>
> A4: We have conducted additional experiments to validate the robustness and efficacy of our Elo-based utility learning algorithm. In the Game of 24 scenario, we tested two variants of RaDAgent to assess the impact of different utility learning approaches:
>
> **Simpler Utility Learning Prompt**: In this variant, we employed a more straightforward utility learning prompt to compare two decision sequences. The prompt is listed as follows:
> ```
> Giving task description and candidate answers, I want you to choose one preferred answer which is more close to success.
>
> *******************************
> {{BEGIN_DESCRIPTION}}
> your_task: {task_description}
> your_query: {input_description}
> {{END_DESCRIPTION}}
> *******************************
>
> *******************************
> {{CANDIDATE_0_START}}
> {candidate_A}
> {{CANDIDATE_0_END}}
> *******************************
> {{CANDIDATE_1_START}}
> {candidate_B}
> {{CANDIDATE_1_END}}
> *******************************
> ```
> And the experimental results is as follows:
>
> | Method  | Success Rate |
> |---------|-----------|
> | DFSDT   | 29.0      |
> | SimplePrompt| 36.0      |
> | RaDAgent| 43.0      |
>
> From the results, we can observe that while there was a decrease in performance with the simpler prompt, RaDAgent still outperforms the best baseline DFSDT. This results highlights a couple of key points:
> - **Impact of Prompt Design**: The experiment demonstrated that the design of the utility learning prompt does indeed impact the performance of the system. A more complex or carefully crafted prompt contributes to better utility assessment, leading to more effective decision-making.
> - **Robustness of Utility Learning**: Despite the reduced performance with a simpler prompt, the fact that RaDAgent continued to outperform the baseline indicates the inherent robustness of our utility learning approach. It suggests that while the prompt design is significant, the core mechanics of our Elo-based utility learning algorithm are strong enough to maintain a competitive edge even under suboptimal conditions.
>
> These findings unveil the need for further research into the optimal design of utility learning prompts. We plan to explore a broader range of prompt complexities and styles to fully understand their impact on the efficacy of the utility learning process in the future.
>
> **Manual-Designed Strategy Comparison**: We also experimented with a manual-designed strategy for comparing decision sequences, instead of LLMs. The strategy is to compare which decision sequence is close to 24 (i.e., the difference between the final calculation result of four number and 24). The closer one wins. If their difference are the same, they tied. The experimental results are as follows:
>
> | Method  | Success Rate |
> |---------|-----------|
> | DFSDT   | 29.0      |
> | Manual  | 26.0      |
> | RaDAgent| 43.0      |
>
> The performance of this variant was notably inferior to that of the standard RaDAgent even DFSDT. This is the manual-designed strategy acts as an external performance measure and it may flawed or even erroneus, which would mislead the decision-making process. This outcome highlights the limitations of manual strategies and affirms the importance of the internal rationality embedded within our RaDAgent for decision-making.

---

### Official Review · Reviewer_Zp4E · 2023-10-31

**Soundness:** 2 fair
**Presentation:** 2 fair
**Contribution:** 2 fair
**Rating:** 5
**Confidence:** 4

**Summary:**

The paper proposes an approach for integrating RL/MDP type methods for use with LLMs. RL methods typically function by interacting with the environment (exploring) and greedily selecting actions (exploiting). However, such methods rely on the Reward function being available and the agent being capable of perceiving rewards at each step.

The authors propose RADAGENT, an agent that can utilize RL-type sequential decision-making methods in the absence of a reward function. In order to do so, the authors devise a customized reward function (reward shaping) that is based on the Elo rating system. Each episode (decision sequence) assigns a reward following three-valued logic (1, 0 or 0.5) depending on the outcome. These are propagated upward in a bottom-up fashion enabling the agent to eventually self-evaluate earlier steps in the decision making process. The exploration process is similar to methods like $\epsilon$-greedy but now conditioned as a softmax over elo scores.

The authors then provide an empirical evaluation on a few datasets and showcase that their approach is able to outperform baseline methods.

**Strengths:**

1. The paper builds on known principles on sequential decision making

2. The results presented have a sizeable performance lead over the baselines

**Weaknesses:**

I think that the paper has a few weaknesses in some key areas that might limit its applicability.

Also, the empirical evaluation section was a bit confusing to read. A bit of reorganization here might help.

1. The paper mentions (Sec 3.2) the initial elo scores for a decision sequence (iteration 1) are fixed. How are comparisons performed? IE how is a "win" determined? I assume it is binary (task completed or not). In this case, are two decision sequences where both are "wins" but one is longer than the other compared? (Longer ones require more API calls I assume).

2. Building upon #2 above, the paper claims that there is a performance measure absent. That would mean that there is no real way of determining the final utility or "win". I think that there is some human-expert knowledge that is required for RADAGENT to be able to function. Could you please clarify this?

3. Why the specific use of Elo and not a different rating system like Glicko? There is not sufficient motivation for the choice of Elo.

**Questions:**

I have asked my questions in the weaknesses section itself.

---

> ### Author Response · Authors · 2023-11-22
>
> Q1: 1. The paper mentions (Sec 3.2) the initial elo scores for a decision sequence (iteration 1) are fixed. How are comparisons performed? IE how is a "win" determined? I assume it is binary (task completed or not). In this case, are two decision sequences where both are "wins" but one is longer than the other compared? (Longer ones require more API calls I assume).
>
> A1: We appreciate the opportunity to clarify these aspects of Elo rating details and comparison determination in the utility learning.
>
> - **How are comparisons performed**: Regarding the comparisons, they are entirely agent-centric, devoid of any external or prior knowledge. This design choice is intentional: **Our goal is to validate the inherent capacity of the LLM-based agent to make rational utility judgments.** You can refer to Appendix A. Specifically, ChatGPT receives two decision sequences (i.e., ChatGPT calling messages) and then judges the better one to return its corresponding index.
> - **How is a "win" determined**: The term 'win' in our context is not a binary indicator to judge whether a decision sequence successfully completes a task. Instead, it assesses which decision sequence demonstrates **superior performance** during the task completion process. Thus, even if two decision sequences both accomplish the task, our method will still judge the better one according to their decision sequence. This comparative approach aims to discern the relative efficacy of different decision sequences, rather than making an absolute determination of task completion. Additionally, the comparison result is triple: A wins B, B wins A, or a Tie, reflecting scenarios where decision sequences A and B exhibit similar performance levels.
> -  **How to compare two that one is longer requiring more API calls**: Our empirical observations indicate that it depends on the specific context of decision sequences. For instance, if a shorter sequence fails while a longer one succeeds, RaDAgent will opt for the longer, successful sequence. Conversely, when both short and long sequences succeed, RaDAgent exhibits a tendency to favor the shorter, more efficient sequence. This preference aligns with intuitive human judgment, reinforcing the notion that LLM-based agents possess the capability to discern and prefer more efficient decision-making pathways.
> - **Initial Elo score is fixed**: The initial Elo score is fixed does not mean the Elo score is fixed during the utility learning process. **The Elo score will be updated through comparison to converge to its accurate value.** Given a new decision sequence, RaDAgent will first assign it with an initial Elo score and then multiple comparisons will be conducted with existing decision sequences to adjust this decision sequence Elo score according to Equation 3 immediately. Hence, the initial Elo score just serves as a standardized starting point for evaluation, ensuring that subsequent Elo score adjustments in an unbiased and consistent manner.

---

> ### Author Response · Authors · 2023-11-22
>
> Q2: The paper claims that there is a performance measure absent. That would mean that there is no real way of determining the final utility or "win". I think that there is some human-expert knowledge that is required for RaDAgent to be able to function. Could you please clarify this?
>
> A2: We acknowledge the inquiry regarding the role of human-expert knowledge in RaDAgent's functionality. It is important to clarify that the integration of such knowledge is seen as complementary to, rather than a replacement for, the agent's inherent rational decision-making capabilities. While expert knowledge can indeed augment the agent’s decision-making process, making it more effective in certain contexts, reliance on such knowledge alone is not without drawbacks.
>
> Primarily, external human expertise can sometimes be limited, flawed, or even erroneous. Additionally, it is impractical to infuse an all-encompassing array of prior knowledge into a general-purpose agent, especially considering the vast and varied nature of real-world tasks. Therefore, in our study, we emphasize that a general-purpose agent like RaDAgent should not solely depend on external human expertise. Instead, it should leverage its built-in rationality to make autonomous decisions, especially in scenarios where reliable external guidance is unavailable or insufficient.
>
> In essence, our approach positions the agent's internal rationality as a fundamental aspect of its decision-making process, while still acknowledging the potential benefits of integrating human expertise where feasible and appropriate. This dual approach ensures that the agent remains adaptable and effective across a wide range of tasks, even in the absence of specific external inputs.

---

> > ### Comment · Reviewer_Zp4E · 2023-11-23
> >
> > Thank you for your responses.
> >
> > I still think this point needs some consideration. I understand that reward shaping etc that provide meaningful signals are not quite required for Radaagent to function but the claim that "there is no performance measure needed" is rather boisterous. There is an inbuilt performance measure of what it means to be rational in the context of whether a decision sequence is better or not.
> >
> > This is the bare minimum signal that is required else the concept of RL and utility functions collapses. I think that claiming that performance measures are absent entirely discounts the fact that the concept of rationality requires at least one signal that differentiates the "good" from the "bad". The evaluation inherently uses such performance measures from LLM's massive training datasets which have utilized this information and thus there are certain assumptions baked in.
> >
> > A clarification that the performance measure like rewards etc that humans need to engineer is not really required would help the clarity of the paper.
> >
> > I thank the authors for their clarifications and will increase my score by 1 point.

---

> ### Author Response · Authors · 2023-11-22
>
> Q3: Why the specific use of Elo and not a different rating system like Glicko? There is not sufficient motivation for the choice of Elo.
>
> A3: Our primary objective in this work was to underscore the significance of internal rationality for the agent in decision-making processes, particularly in evaluating the utility of each choice to determine the optimal one. In this context, the crucial factor was to employ a method that could provide a quantitative assessment of utility through a straightforward yet robust mechanism. We opted for the Elo rating system due to its simplicity and proven effectiveness, especially in environments where pairwise comparisons are pivotal. This system's straightforward nature aligns well with our objective to demonstrate the agent's capability to internalize utility judgment without overly complex calculations or dependencies. Our experimental results affirm that the Elo rating algorithm successfully showcases the effectiveness of internalizing utility judgment in the decision-making process.
>
> While Elo was chosen for its current applicability, the reviewer's suggestion of exploring alternative rating systems like Glicko is indeed valuable. We plan to consider this insightful direction in our future research to further enhance our understanding of rational decision-making in the agent.

---

> ### Author Response · Authors · 2023-11-22
>
> Q4: How much hyperparameter tuning was done on the hyperparameters $\hat{d}$ and $K$? What was the performance of other hyperparameters that were tried?
>
> A4: For the initial Elo score of $\hat{d}$, we want to clarify the role of the initial Elo score in our RaDAgent.
>
> First, it is crucial to understand that in the Elo rating algorithm, the decision-making process is influenced by the relative difference between Elo scores of decision steps, rather than their absolute values. You can refer to Equation 1 in our paper.
>
> Second, the essence of the Elo rating system lies in its dynamic nature, where scores are adjusted based on pairwise comparisons over time. This means that regardless of the initial score assigned to each decision step, the subsequent adjustments made through pairwise comparisons are what determine the final, accurate assessment of each decision's utility. Therefore, the initial score primarily serves as a starting point, and its specific value is not critical to the overall decision-making process.
>
> Consequently, the initial Elo score assigned to decision steps does not fundamentally impact the outcome of the comparisons.
>
> For the hyperparameter $K$, we conducted a series of experiments with different values of $K=10$, $K=50$, and $K=100$ on the Game of 24 scenario. The experimental results are listed as follows:
>
> | Method  | Succeed Rate |
> |---------|-----------|
> | $K=10$    | 34.0    |
> | $K=50$    | 48.0      |
> | $K=100$   | 43.0      |
>
> Through these experiments, we observed that $K=50$ yielded the most optimal performance for our RaDAgent model. It is important to note that $K$ in the Elo update algorithm functions analogously to the learning rate in Stochastic Gradient Descent optimization algorithms. The choice of $K$ significantly influences the rate at which the Elo scores converge to their accurate values. A larger $K$ may lead to instability in the Elo scores, as it causes larger adjustments, thereby potentially overshooting the optimal value. Conversely, a smaller $K$ can result in slower convergence, necessitating more comparisons to reach an accurate assessment. We acknowledge that further study is warranted to refine the selection of $K$ for different scenarios and environments.
>
> To further assess the impact of these hyperparameters, we plan to conduct additional experiments in the future. We appreciate your valuable suggestions and will incorporate them into our future work.

---

### Official Review · Reviewer_ynxx · 2023-10-31

**Soundness:** 3 good
**Presentation:** 3 good
**Contribution:** 3 good
**Rating:** 8
**Confidence:** 4

**Summary:**

The submission proposes a new approach to LLM-based decision making that it calls RaDAgent. The approach works by computing elo scores to decision sequences using the outcomes of LLM-based pairwise evaluations of decision sequences.

**Strengths:**

I am not aware of any existing approaches resembling the one proposed in the paper. The key idea of backing out elo rankings using pairwise evaluations from an LLM is intuitively appealing, but subtle enough that the submission merits credit for originality.

The submission is mostly clear and mostly well-written.

LLM decision making is an important problem and the submission's results are strong relative to existing previous works.

**Weaknesses:**

### Clarity Issues

Below I list a couple of places I had trouble following along with some explanations.

---

> In contrast, RADAGENT assigns lower scores to fewer potential decision steps, displaying a trend for exploring novel avenues, which exemplifies a scenario demanding diversity in exploration.

I wasn't able to parse this sentence.

---

In RQ5, I feel there could be some more guidance to the reader. It seems like RaDAgent has the highest or tied for highest incidence ratio for both Hallucinated Tool and Tool Call Error. If I understand the meaning of incidence ratio correctly, high values are undesirable. Yet the text does not really provide any discussion of these results, so I am left a bit confused as to what to make of them. Is this because the other methods fail for other reasons? What do these numbers really tell us?

### Structure of Paper

I think related work would be better placed earlier in the paper (prior to experiments). It would help the reader better contextualize the both RaDAgent and the baselines.

I also find the structure of the experiments is a bit rambly. Having section an experimental settings section and RQ1-5 is fine, but then having an additional discussion section after 5 previous subsections that are already discussing results feels kind of disorganized, especially since there is some overlap. In particular, each section of the discussion contrasts RaDAgent against existing methods, which according to the description of RQ5 (What are the key differentiating factors of RADAGENT against other methods?), is supposed to be within its purview.

**Questions:**

How important is the utility learning prompt to performance? Were other prompts tried? How much worse would the performance be with a simpler prompt?

How much hyperparameter tuning was done on the hyperparameters $\hat{d}$ and $K$? What was the performance of other hyperparameters that were tried?

---

Overall, I liked the submission. The idea to use elo rankings from pairwise comparison makes intuitive sense to me and the results seem strong.

---

> ### Author Response · Authors · 2023-11-22
>
> Q1: In RQ5, I feel there could be some more guidance to the reader. It seems like RaDAgent has the highest or tied for highest incidence ratio for both Hallucinated Tool and Tool Call Error. If I understand the meaning of incidence ratio correctly, high values are undesirable. Yet the text does not really provide any discussion of these results, so I am left a bit confused as to what to make of them. Is this because the other methods fail for other reasons? What do these numbers really tell us?
>
> A1: It is essential to clarify that the incidence ratios for Hallucinated Tool and Tool Call Error are calculated based on the entire exploration process, encompassing the full breadth of the explored decision tree. As RaDAgent conducts a diverse and extensive exploration, it will experience more errors, causing a higher incidence ratio. This diverse exploration is integral as it allows RaDAgent to thoroughly evaluate a wide array of possible decision pathways, even those that are less conventional or more prone to errors.
>
> More importantly, we have examined the fix ratio which demonstrates the performance of each method to fix the Hallucinated Tool and Tool Call Error to accomplish the task finally. Despite exploring riskier or more error-prone decisions, RaDAgent effectively utilizes the Elo-based utility learning mechanism to learn from diverse explorations and subsequently pinpoint the most efficient and error-free pathway. The high fix ratio underlines RaDAgent's ability to rectify potential errors encountered during the exploration phase, ultimately leading to a reliable and effective decision-making process.
>
> We will further clarify these points in the revised paper to provide a better understanding of the incidence ratio's significance and the robustness of RaDAgent's decision-making process.

---

> ### Author Response · Authors · 2023-11-22
>
> Q2: I wasn't able to parse this sentence: "In contrast, RaDAgent assigns lower scores to fewer potential decision steps, displaying a trend for exploring novel avenues, which exemplifies a scenario demanding diversity in exploration."
>
> A2: Apologize for our unclear expression and we would like to offer a revision for clarity.
>
> The original sentence intended to describe RaDAgent's exploration mechanism, wherein it assigns lower Elo scores to decision steps that it assesses as less likely to contribute to successful task resolution. This scoring framework is designed to steer the agent away from investing computational resources in exploring decision steps that are unlikely to be valuable. As a result, RaDAgent is guided to investigate a broader spectrum of potential decisions, which inherently leads to a more diverse exploration pattern.
>
> To express this more clearly, we will revise the sentence to read:
> ```
> RaDAgent is designed to discount the less promising decision steps by assigning them lower Elo scores, thereby encouraging the exploration of other potential and more promising decisions. This fosters a wide-ranging search across the decision space, as opposed to a localized search around less advantageous steps, leading to a richer and more diverse set of explored possibilities.
> ```
>
> We believe this revised statement more accurately conveys the underlying principle of RaDAgent's exploration strategy and its commitment to diverse exploration by avoiding less fruitful paths.

---

> ### Author Response · Authors · 2023-11-22
>
> Q3: Structure of Paper about Related Work and Discussion.
>
> A3: We gratefully thank you for your insightful suggestions to enhance the structural coherence of our paper. In response, we will revise the structure of our manuscript in the final version. First, the Related Work section will be advanced to follow the introduction, which will provide our readers with a foundational understanding of the existing landscape against which our contributions can be more meaningfully assessed. Second, we will integrate the Discussion section into each corresponding section of the experiments. This integration will eliminate redundancies and provide immediate insights into the significance and implications of our results, as they are presented. We extend our thanks for this constructive critique and the opportunity to refine our work.

---

> ### Author Response · Authors · 2023-11-22
>
> Q5: How important is the utility learning prompt to performance? Were other prompts tried? How much worse would the performance be with a simpler prompt?
>
> A5: We appreciate the opportunity to discuss the importance of our model's performance with respect to the utility learning prompt. To this end, we conducted comparative experiments on the Game of 24 setting using a simpler utility learning prompt as follows:
> ```
> Giving task description and candidate answers, I want you to choose one preferred answer which is more close to success.
>
> *******************************
> {{BEGIN_DESCRIPTION}}
> your_task: {task_description}
> your_query: {input_description}
> {{END_DESCRIPTION}}
> *******************************
>
> *******************************
> {{CANDIDATE_0_START}}
> {candidate_A}
> {{CANDIDATE_0_END}}
> *******************************
> {{CANDIDATE_1_START}}
> {candidate_B}
> {{CANDIDATE_1_END}}
> *******************************
> ```
> The experimental results are as follows:
>
> | Method  | Succeed Rate |
> |---------|-----------|
> | DFSDT   | 29.0      |
> | SimplePrompt  | 36.0      |
> | RaDAgent| 43.0      |
>
> From the results, we can observe that while there was a decrease in performance with the simpler prompt, RaDAgent still outperforms the best baseline DFSDT. This results highlights a couple of key points:
> - **Impact of Prompt Design**: The experiment demonstrated that the design of the utility learning prompt does indeed impact the performance of the system. A more complex or carefully crafted prompt contributes to better utility assessment, leading to more effective decision-making.
> - **Robustness of Utility Learning**: Despite the reduced performance with a simpler prompt, the fact that RaDAgent continued to outperform the baseline indicates the inherent robustness of our utility learning approach. It suggests that while the prompt design is significant, the core mechanics of our Elo-based utility learning algorithm are strong enough to maintain a competitive edge even under suboptimal conditions.
>
> These findings unveil the need for further research into the optimal design of utility learning prompts. We plan to explore a broader range of prompt complexities and styles to fully understand their impact on the efficacy of the utility learning process in the future.

---

### Meta-Review · Area_Chair_o1GN · 2023-12-05

**Metareview:**

I am not certain what the right decision should be. This is a submission I wish to discuss further with the SAC. The submission received border line scores. I personally do not find it to be very novel, other than applying existing ideas to an LLM. Concretely, the authors implemented planning using ELO scores on top of a language agent. It is less clear what the new method has to offer when compared to more traditional planning methods, like MCTS.

**Justification For Why Not Higher Score:**

Border line scores and limited contributions.

**Justification For Why Not Lower Score:**

n/a

---

### Decision · Program_Chairs · 2024-01-16

Reject